# a-Synuclein and lipids in erythrocytes of Gaucher disease carriers and patients before and after enzyme replacement therapy

Marina Moraitou[1], Georgios Sotiroudis[2], Nikolaos Papagiannakis[3], Maria M. J. Ferraz[4], Aristotelis Xenakis[2], Johannes M. F. G. Aerts[4], Leonidas Stefanis[3,5], Helen Michelakakis[1] *

1 Department of Enzymology and Cellular Function, Institute of Child Health, Athens, Greece, 2 Institute of Chemical Biology, National Hellenic Research Foundation, Athens, Greece, 3 1st Department of Neurology, Eginition Hospital, Medical School, National and Kapodistrian University of Athens, Athens, Greece, 4 Leiden Institute of Chemistry, Leiden University, Leiden, The Netherlands, 5 Laboratory of Neurodegenerative Diseases, Biomedical Research Foundation of the Academy of Athens, Athens, Greece

* ecfdept@ich.gr

**Data Availability Statement:** All relevant data are within the paper and its Supporting Information files.

## Abstract

It is well established that patients with Gaucher disease, as well as carriers of the disease have an increased risk for developing Parkinson's disease. A plethora of evidence suggests that disturbed α-Synuclein homeostasis is the link between Gaucher disease and Parkinson's disease. The pathogenic mechanism linking these entities is still a topic of debate and both gain- and loss-of-function theories have been put forward, which however are not mutually exclusive. In the present study we expanded our previous studies to include not only Gaucher disease patients but also Gaucher disease carriers and Gaucher disease patients following Enzyme Replacement Therapy. In these groups we investigated α-Synuclein in red blood cell membranes in association with lipid abnormalities described in Gaucher disease. These included glucosylceramide and its species, glucosylsphingosine, glucosylcholesterol and plasmalogens. Increased oligomerization of α-Synuclein in red blood cell membranes was observed not only in Gaucher disease patients but also in carriers of the disease. There were no qualitative differences in the lipids identified in the groups studied. However, significant quantitative differences compared to controls were observed in Gaucher disease patients but not in Gaucher disease carriers. Enzyme Replacement Therapy reversed the biochemical defects and normalized α-Synuclein homeostasis, providing for the first time evidence in human subjects that such homeostatic dysregulation is reversible. Further studies investigating α-Synuclein status during the differentiation of erythroid progenitors could provide new data on the pathogenic mechanism of α-Synuclein oligomerization in this system.

## Introduction

Gaucher disease (GD) is a rare autosomal recessive disorder that belongs to the group of lysosomal storage diseases (Mendelian Inheritance in Man (MIM) 230800, 230900 and 231000). It

**Funding:** The authors received no specific funding for this work.

**Competing interests:** The authors have declared that no competing interests exist.

results from the functional deficiency of the lysosomal enzyme β-glucocerebrosidase (GCase; also called glucosylceramidase or acid β-glucosidase, E.C. 3.2.1.45) which, in most instances, is associated with mutations occurring in the β-glucocerebrosidase gene (*GBA1*; MIM #606463, GenBank accession no. J03059.1). The deficient enzyme activity leads to the accumulation of its substrate glucosylceramide (GlcCer), mainly in tissue macrophages, transforming them into the characteristic Gaucher cells. Along with the accumulation of GlcCer several other lipid abnormalities have also been described in patients with GD [1–3].

Parkinson's disease (PD), a multifactorial neurodegenerative disorder, is the second most common neurodegenerative disorder with a prevalence of 1% in the population older than 60 years [4].

The pathological hallmark of both familial and sporadic PD is the progressive degeneration of dopaminergic neurons in the substantia nigra and other brain regions, accompanied by the accumulation of intracellular protein inclusions known as Lewy bodies. These inclusions are positive for α-Synuclein (α-Syn) and ubiquitin [5]. α-Syn is an abundant, highly conserved neuronal presynaptic protein of 140 amino acids. Its misfolding, oligomerization and fibrilization, collectively termed aggregation, participates in neurodegeneration in PD and other synucleinopathies [6]. α-Syn is a lipid binding protein and its interaction with lipid membranes is important in the aggregation process, while it may also affect the properties of the membranes [7, 8].

The presence of mutations in the *GBA1* gene is recognized as the main genetic risk factor for the development of PD. This association was first recognized in the clinic with the diagnosis of PD in GD patients and their obligate carrier family members [9–12]. This initial observation was subsequently validated and substantiated through neuropathological studies in such patients, further family studies of GD patients, as well as the investigation of *GBA1* mutations in patients with PD [13–19]. In particular the neuropathological evaluation of brains from GD patients with parkinsonism and PD subjects who carry *GBA1* mutations along with studies in animal models of GD disease, neuronal cultures with deficient GCase activity, human iPSC neurons derived from *GBA1* mutation carriers, as well as induced pluripotent stem cell (iPSC)-derived neurons from GD and PD individuals carrying *GBA1* mutations, have revealed the presence of a-Syn aggregates or accumulation [17, 20–23]. These findings show that *GBA1* mutation carriers exhibit the typical hallmarks of PD and suggest a pathophysiological link between mutant glucocerebrosidase expression and a-Syn metabolism. However, the pathogenic mechanism linking these entities is still a topic of debate and both gain- and loss-of-function theories have been put forward. Loss of GCase activity and the subsequent lipid abnormalities, direct interaction between mutant GCase and α-Syn molecules, triggering of the unfolded protein response by mutant GCase protein, impaired lysosomal function, disturbed autophagy have all been proposed as possible underlying mechanisms linking these entities [22–26].

It has been shown that α-Syn is abundantly expressed in erythroid cells, including erythroblasts, reticulocytes and erythrocytes, in both the bone marrow and the peripheral blood, where α-Syn is localized in both the cytoplasm and the plasma membrane of circulating erythrocytes [27]. Several studies, including from our own group, have shown that increased oligomerization of α-Syn is observed in red blood cell (RBC) membranes and plasma from GD patients [28–34]. Furthermore, we have shown that in GD the α-Syn dimer/monomer ratio positively correlated with the levels of GlcCer, the glucosylceramide/ceramide (GlcCer/Cer) ratio and negatively with the levels of malonyldialdehyde (MDA) and plasmalogens [32].

In the present study we expanded our previous studies to include not only GD patients but also GD carriers and GD patients following Enzyme Replacement Therapy (ERT). In these groups we investigated α-Syn in RBC membranes in association with lipid abnormalities described in GD disease. These included GlcCer and its species, hexosylsphingosine (HexSph), being largely glucosylsphingosine (GlcSph), glucosylcholesterol (GlcChol) and plasmalogens.

## Subjects and methods

### Subjects

A total of 113 individuals, all of Caucasian origin, were included in this study. They were subdivided in 5 different groups.

Group A (GrA) included a total of 45 patients with GD (type I n: 40, type II n: 2, type III n: 3). Group A1 (GrA1) included 13 of the GrA patients (type I n: 12 and type III n: 1) prior to the initiation of ERT, while Group B (GrB) included these 13 patients following one year of ERT. The remaining 32 patients in group A either were not treated or there was no blood sample available after one year of treatment. Group C (GrC) included 19 GD carriers. A total of 49 control individuals (Group D; GrD) were also studied. The demographics of all the individuals studied are shown in Table 1. The genotypes of the GD patients and carriers studied are shown in S1 Table. The individuals included in the study had no reported signs of PD at the time of the investigation. The protocol of the study was approved by the Ethics and Scientific Committees of Eginition Hospital of the National and Kapodistrian University of Athens. The need for consent was waived by the ethics committee.

## Materials and methods

RBCs were prepared from heparinized blood samples through spinning at 2,200 x g for 10 min at 4˚C and three consecutive washing steps with one volume of 0.9% NaCl. The RBC pellets were stored in aliquots at -80˚C in 0.9% NaCl (1:1, v/v) and were thawed (cell lysis) upon use. The number of RBCs per μL was counted in an aliquot of the cell suspension prior to freezing.

**Table 1. Demographics of the individuals studied.**

| Groups | Age | | | Gender |
|---|---|---|---|---|
| **GrA**<br>n = 45 | **Type I**<br>n = 40<br>range: 6–69 yrs | **All types**<br>range: 5 d-69 yrs<br>median: 25 yrs | | **All types**<br>male: 23<br>female: 22 |
| | **Type II**<br>n = 2<br>range: 5 d-5 mo | | | |
| | **Type III**<br>n = 3<br>range: 2–8 yrs | | | |
| **GrA1**<br>n = 13 | **Type I**<br>n = 12<br>range: 6–65 yrs | range: 2–65 yrs<br>median: 21 yrs | | male: 4<br>female: 9 |
| | **Type III**<br>n = 1<br>2 yrs | | | |
| **GrB**<br>n = 13 | **Type I**<br>n = 12<br>range: 7–66 yrs | range: 3–66 yrs<br>median: 22 yrs | | male: 4<br>female: 9 |
| | **Type III**<br>n = 1<br>3yrs | | | |
| **GrC**<br>n = 19 | range: 1 mo-64 yrs<br>median: 41 yrs | | | male: 6<br>female: 13 |
| **GrD**<br>n = 49 | range: 4–76 yrs<br>median: 38 yrs | | | male: 20<br>female: 29 |

GrA, Gaucher disease patients receiving no treatment; GrA1, subgroup of GrA before ERT; GrB, GrA1 patients following one year of ERT; GrC, Gaucher disease carriers; GrD, controls

Plasmalogen levels were measured by gas chromatography (7890A, Agilent, USA) in a lipid extract of RBC membranes after methyl transesterification with methanolic HCl (Supelco). The α, β ether bond in the plasmalogen molecules was thus converted to the dimethylacetal derivative of the corresponding aldehyde (DMA) [3]. The extracted lipids from 50 μL of RBC lysates were finally dissolved in 60 μl hexane and 1 μL was injected into the column (HP-FFAP, Agilent, USA) [35]. Plasmalogen levels were expressed as the ratio of the C16:0 DMA to methylpalmitate (C16:0) and C18:0 DMA to methylstearate (C18:0).

GlcCer isoforms were isolated from 100 μL of RBC lysates, as described by Boutin et al [36], by solid phase extraction using the hydrophilic-lipophilic balance cartridges (OASIS, Waters Corp USA) with deuterated N-palmitoyl-D3-glucosylceramide (D3C16:0-GlcCer) as internal standard and N-pentadecanoyl-galactosylceramide (C15:0-GalCer) as calibration curve standard (both from Matreya LCC, Pleasant Gap, USA). The extracted lipids were evaporated to dryness and were resuspended in 100 μL acetonitrile 95.5% / methanol 2.5% / water 2% / formic acid 0.5% /ammonium acetate 5mM (mobile phase of the following high pressure liquid chromatography (HPLC) method) for analysis.

The quantification of the GlcCer isoforms was performed by LC-MS/MS using a modification of the method of Boutin et al [36]. Instrumentation consisted of a 3200Q TRAP triple–quadrupole linear ion trap mass spectrometer fitted with a Turbolon Spray interface (SCIEX, USA) and an Agilent 1200 HPLC system consisting of a G1379 B micro vacuum degasser, a 1312A binary pump, a G1329 autosampler and a G1316A column compartment (Agilent, USA).

The separation of the GlcCer isoforms was performed with an Agilent ZORBAX HILIC Plus column (50 mm x 2.1 mm inner diameter, 3.5 μm particle size) with RRLC inline filter kit (2.1 mm, 0.2 μm filter) (Agilent, USA). The isocratic mobile phase consisted of acetonitrile 95.5% / methanol 2.5% / water 2% / formic acid 0.5% / ammonium acetate 5mM. The flow rate was set at 150 μL/min and the injection volume was 5 μL. Electrospray ionization operating in positive mode was used for all analytes. Compound-specific optimization of MS/MS parameters was performed via direct infusion of a mixture of standards reference solution (0.5 μg/mL each in the mobile phase used for the HPLC) using a syringe pump. Source parameters were set to optimal values after flow injection analysis (FIA) source optimization. Quadrupoles one and three were set to unit resolution. Data acquisition was performed in the multi reaction monitoring (MRM) mode and the five MRM reactions analyzed corresponded to the calibration curve standard C15:0-GalCer (m/z 686.5), the internal standard D3C16:0-GlcCer (m/z 703.6) and the three GlcCer's identified: N-palmitoyl-glucosylceramide (C16:0-GlcCer; m/z 700.5), N-stearoyl-glucosylceramide (C18:0-GlcCer;m/z 728.6) and N-nervonoyl-glucosylceramide (C24:1-GlcCer;m/z 810.7). The same fragment ion (m/z 264.5) corresponding to the dehydrated sphingosine moiety, was monitored for all the molecules analyzed and dwell time was set to 200 msec. Source parameters were set as follows: Curtain Gas (CUR), 20; Temperature (TEM), 400; CAD gas, 5; Gas 1 (GS1), 40; Gas 2 (GS2), 40; Ion Spray (IS), 4500. The data were processed using the Analyst Software program (version 1.4.2) (SCIEX USA).

The quantification of glycosylated sphingosine (HexSph, the total of glucosylsphingosine and galactosylsphingosine) and of glycosylated cholesterol (HexChol, the total of glucosylcholesterol (GlcChol) and galactosylcholesterol (GalChol) was performed by reverse-phase liquid chromatography using a Waters UPLC-Xevo-TQS micro and a BEH C18 column (2.1 × 50 mm with 1.7 μm particle size, Waters) at 23˚C. Mass spectrometry detection in positive mode using an electrospray ionization (ESI) source was carried out with a Xevo TQS micro instrument. Data were analyzed with Masslynx 4.1 Software (Waters Corporation) [37]. LC-MS-grade methanol, water, formic acid and HPLC-grade chloroform were purchased from Biosolve. All remaining reagents were obtained from Merck, unless otherwise stated in the text.

The internal standards 13C6-GlcChol and 13C5-GlcSph were synthesized at the Department of Bio-organic Synthesis of the Leiden Institute of Chemistry, Leiden University.

The lipids were extracted from 20 μL of sample by a modified Bligh and Dyer extraction using acidic buffer (100 mM ammonium formate buffer pH 3.1) with 13C5-GlcSph and 13C6-GlcChol (0.1 pmol/μL) as internal standards. To this, 300 μL methanol and 150 μL chloroform were added, stirred and centrifuged for 10 min at $15,700 \times g$ to precipitate protein. The supernatant was transferred to a clean tube, 150 μL chloroform and 270 μL buffer were added to promote phase separation. The upper phase was transferred to a clean tube (extract A). To the lower phase, 300 μL methanol and 270 μL buffer were added and stirred. The upper phase was pooled with extract A. This was dried at 45˚C in an Eppendorf Concentrator Plus and further extracted with butanol/H2O (1:1). The butanol phase was taken to dryness. The residue was dissolved in 100 μL MeOH, stirred and sonicated in a bath for 30 s and centrifuged for 5 min at $15,700 \times g$. Finally, 10 μL of the solution was applied to the UPLC-MS. The lower phase of the extraction procedure was transferred to an Eppendorf tube, dried at 45˚C and followed by a butanol/H2O extraction as described above. Finally, the sample was resuspended in 100 μL MeOH and 10 μL of the solution was applied to the UPLC-MS.

In order to separate galactose- from glucose-containing molecules, samples were run in an ethylene bridged hybrid (BEH) hydrophilic interaction liquid chromatography (HILIC) column as previously described [38]. Glycosylated sphingosine and glycosylated cholesterol were extracted using the acidic Bligh and Dyer procedure previously described, with lipids resuspended in acetonitrile:methanol (9:1, v/v) prior to transfer to LC-MS vials. LC-MS/MS measurements were performed using a Waters UPLC-Xevo-TQS micro instrument (Waters, Corporation, Milford, USA) in positive mode using an ESI source. A BEH HILIC column (2.1 × 100 mm with 1.7 μm particle size, Waters) was used at 30˚C as described before [38]. Eluent A contained 10 mM ammonium formate in acetonitrile/water (97:3, v/v) with 0.01% (v/v) formic acid, and eluent B consisted of 10 mM ammonium formate in acetonitrile/water (75:25, v/v) with 0.01% (v/v) formic acid. HexSph was eluted in 18 min with a flow of 0.4 ml/min using the following program: 85% A from 0 to 2 min, 85–70% A from 2 to 2.5 min, 70% A from 2.5 to 5.5 min, 70–60% A from 5.5 to 6 min, 60% A from 6 to 8 min, 60–0% A from 8 to 8.5 min, 0–85% A from 8.5 to 9.5 min, and re-equilibration of the column with 85% A from 10 to 18 min. HexChol was eluted in 18 min with a flow of 0.25 ml/min using the following program: 100% A from 0 to 3 min, 100–0% A from 3 to 3.5 min, 0% A from 3.5 to 4.5 min, 0–100% A from 4.5 to 5 min, and re-equilibration with 100% A from 5 to 18 min. Data were analysed with MassLynx 4.1 software (Waters Corporation).

Levels of monomeric and dimeric α-Syn in RBC membranes were assessed using Western Immunoblotting. Red blood cell membrane pellets were purified from lysed RBCs through repeated washes in cold PBS and centrifugation at 1,000 g for 10 min at 4˚C. Supernatants containing cytosolic proteins, mainly hemoglobin, were discarded. The membrane pellets were then solubilized with 50 μl STET lysis buffer (50 mM Tris (pH 7.6), 150 mM NaCl, 2 mM EDTA, 1% Triton-X) on ice for 30 min as described in Argyriou et al [31]. Forty micrograms of protein were run on 8% sodium dodecyl sulfate-polyacrylamide gels. Blots were probed with the antibodies directed against α-synuclein (C-20 polyclonal antibody, sc-7011-R, Santa Cruz Biotechnology) and glyceraldehyde 3-phosphate dehydrogenase (GAPDH) (AB2302, Millipore). The ratio of monomeric or dimeric α-synuclein to GAPDH, used as loading control, was first assessed. To make comparisons between different experiments, an internal control was always included on each blot as the reference point [29]. Ratios of dimer to monomer of α-Syn were assessed in a similar fashion. Values were normalized using the same blood sample as an internal control, the dimer to monomer ratio of which was set as 1.

## Statistical analysis

Statistical analysis was performed using R version 4.1 with RStudio. All data underwent a normality test (Shapiro–Wilk) and was found to be non-normally distributed. The statistical significance of the differences between median concentration levels in the different groups was assessed with Kruskal-Wallis test, with Dunn's post hoc test. Wilcoxon's signed-rank test was utilized for checking the difference between pre- (GrA1) and post-treatment (GrB) α-Syn and lipid levels. Spearman's method was used for correlation studies. p-values less than 0.05 were considered significant.

## Results

The GlcCer species detected in the RBC membranes of GD patients before (GrA) and after treatment (GrB), carriers of the disease (GrC) and control individuals (GrD) were C16:0-GlcCer, C18:0-GlcCer and C24:1-GlcCer. In all groups studied C16:0-GlcCer was the main GlcCer species identified.

The use of a HILIC column allows the separation of glucosyl- and galactosyl-lipids. Analysis of six samples revealed that in the samples studied, the HexSph was on average 90% GlcSph (S2 Table). On the other hand, no galactosylcholesterol was detected in any of the samples.

Representative immunoblots of α-synuclein monomer and dimer for the four different study groups are shown in Fig 1.

According to our results a statistically significant increase in all the GlcCer species, as well as their sum (sum-GlcCer), was observed in GD patients compared to control individuals (C16:0-GlcCer: $p < 0.001$; C18:0-GlcCer: $p < 0.001$; C24:1-GlcCer: p = 0.027; sum-GlcCer: $p < 0.001$) (Fig 2, S3 Table). Similar results were obtained when GD patients were compared to GD carriers. A statistically significant increase in all the above parameters was observed in GD patients compared to carriers (C16:0-GlcCer: $p < 0.001$; C18:0-GlcCer: $p < 0.001$; C24:1-GlcCer:

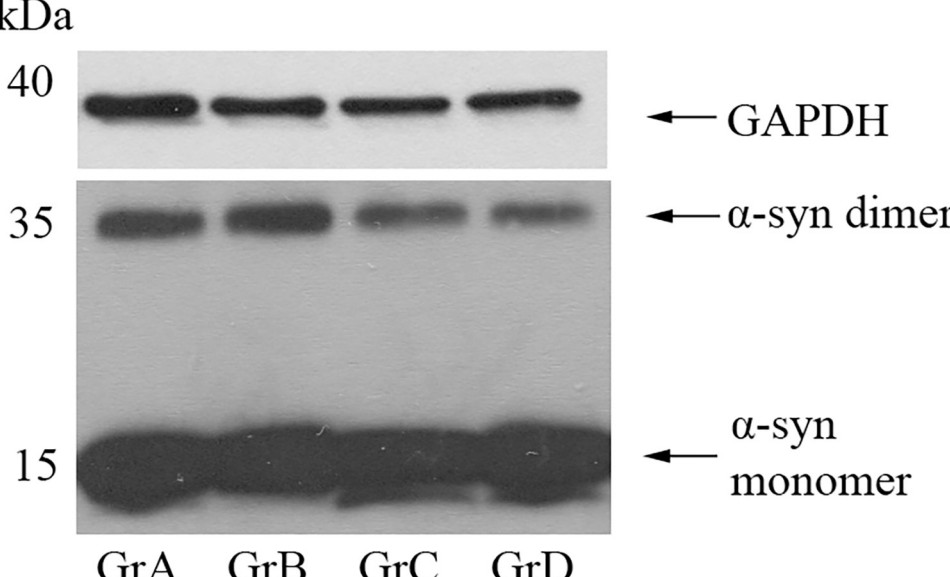

**Fig 1. Representative immunoblots of α-synuclein monomer and dimer for the four different study groups.** GrA, Gaucher disease patients receiving no treatment; GrB, Gaucher disease patients after treatment; GrC, Gaucher disease carriers; GrD, controls. Quantification was made relative to the levels of the loading control (GAPDH, glyceraldehyde 3-phosphate dehydrogenase).

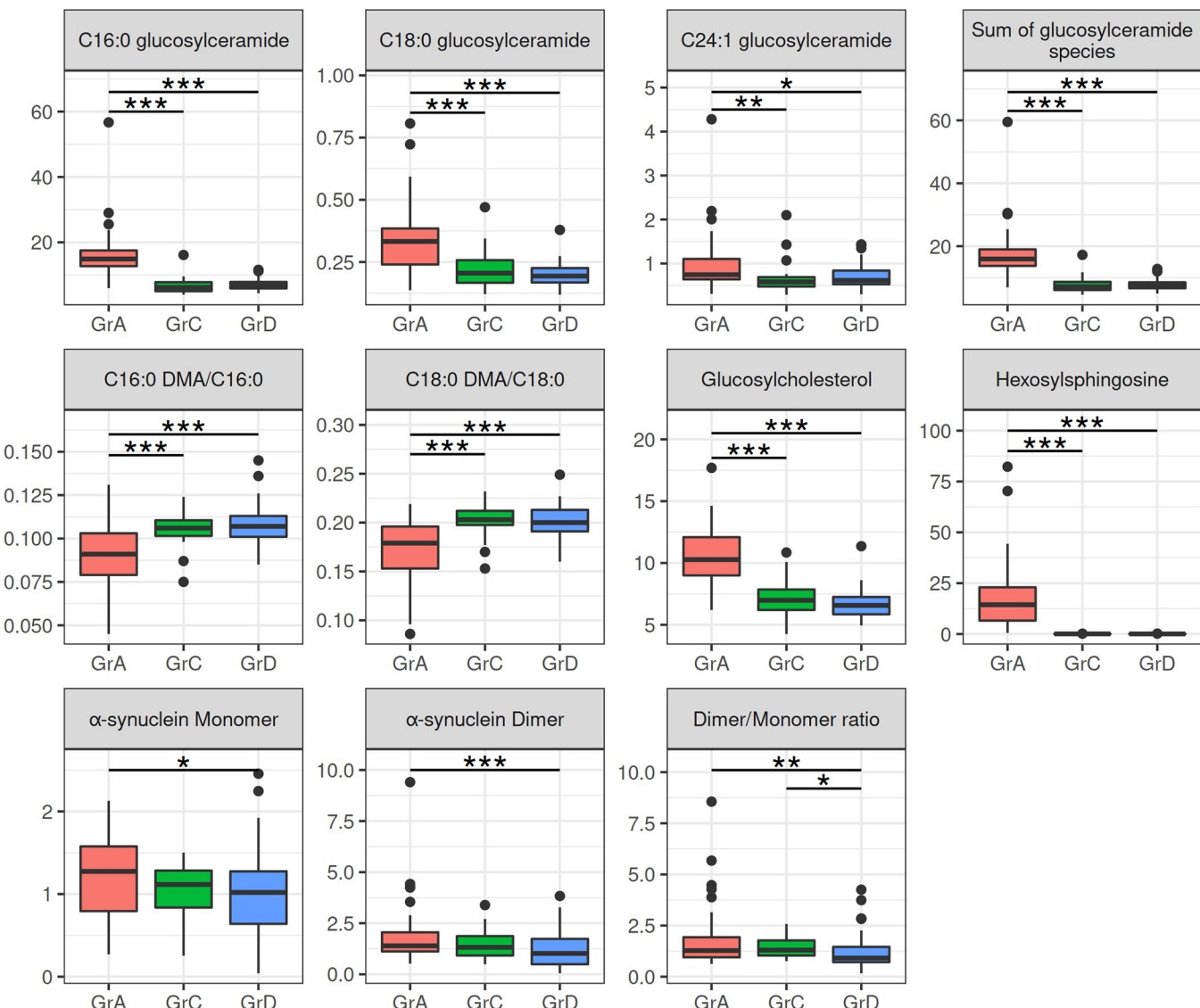

**Fig 2. Red blood cell membrane levels and statistical comparison of the lipids and of α-Synuclein species studied in Gaucher disease patients receiving no treatment, Gaucher disease carriers and controls.** C16:0-glucosylceramide: N-palmitoyl-glucosylceramide; C18:0-glucosylceramide: N-stearoyl-glucosylceramide; C24:1-glucosylceramide: N-nervonoyl-glucosylceramide; C16:0 DMA/C16:0, C16:0-plasmalogens; C18:0 DMA/C18:0, C18:0-plasmalogens; GrA, Gaucher disease patients receiving no treatment; GrC, Gaucher disease carriers; GrD, controls. Y-axis: the C16:0-, C18:0- and C24:1- glucosylceramide species, their sum, hexosylsphingosine and glucosylcholesterol are expressed in pmoles/$10^8$cells. The C16:0- and C18:0-plasmalogens are expressed as the ratios of C16:0 DMA to C16:0 and C18:0 DMA to C18:0, respectively. The α-synuclein monomer and dimer are expressed as the ratio of monomeric and dimeric α-Syn to GAPDH, respectively. The α-synuclein dimer/monomer ratio is expressed as the ratio of dimeric to monomeric α-Syn normalized against a reference sample, the dimer to monomer ratio of which was set as 1. $^*$p<0.050, $^{**}$p<0.010, $^{***}$p< 0.001. Data were analyzed using Kruskal-Wallis test, with Dunn's post hoc test.

p = 0.007; sum-GlcCer: p<0.001). On the other hand, no statistically significant difference in any of the GlcCer species or their sum was observed between GD carriers and control individuals. GlcChol and HexSph in varying amounts were detected in RBC membranes of all the groups studied (Fig 2, S3 Table). GD patients, compared to carriers and control individuals, showed a statistically significant increase (p<0.001) in both GlcChol and HexSph levels. No statistically significant differences in either GlcChol or HexSph levels were observed between GD carriers and control individuals.

RBC plasmalogen levels were estimated as their DMA derivatives and their relative amounts were expressed as the ratio between C16:0 DMA and C16:0, as well as C18:0 DMA and C18:0 GD patients, compared to carriers and control individuals, showed a statistically significant reduction in the levels of both plasmalogen species (p<0.001). No statistically significant differences in either plasmalogen species was observed between GD carriers and control individuals (Fig 2, S3 Table).

A statistically significant increase in α-Syn monomer (p = 0.023), dimer (p = 0.001) levels and α-Syn dimer/monomer ratio (p = 0.005) was observed in GD patients compared to controls. Although an increase in α-Syn dimer levels was observed in GD carriers (median: 1.32; range: 0.49–3.39) compared to controls (median: 1.02; range: 0.05–3.83) this difference did not reach statistical significance. A statistically significant increase was however observed in the α-Syn dimer/monomer ratio (p = 0.019) in GD carriers compared to controls. On the other hand no statistically significant differences were observed between GD patients and carriers (Fig 2, S4 Table).

One year of ERT resulted in a statistically significant reduction in C16:0-GlcCer (p<0.001) and the sum-GlcCer (p = 0.006) in the treated patients (GrB) compared to their pretreatment status (GrA1). Although not statistically significant, a reduction was also observed in the median levels of C18:0-GlcCer and C24:1-GlcCer following treatment. Furthermore, treatment was also associated with significant reductions in GlcChol (p<0.001), HexSph (p<0.001) levels and the α-Syn dimer/monomer ratio (p = 0.011) and in a statistically significant increase in C16:0 DMA/C16:0 (p = 0.036) but not of C18:0 DMA/C18:0. Treated patients, when compared to controls, showed significant increases only in GlcChol (p = 0.006) and HexSph (p<0.001) with no significant differences in any of the other parameters (Fig 3, S5 and S6 Tables).

The correlation between α-Syn and the lipids studied was investigated. In GD patients receiving no treatment a significant positive correlation was observed between the α-Syn dimer/monomer ratio and C16:0-GlcCer (rs = 0.363; p = 0.015), C18:0-GlcCer (rs = 0.332; p = 0.026), as well as the sum-GlcCer (rs = 0.371; p = 0.013). Furthermore, a significant positive correlation was also observed in this group between the α-Syn dimer levels and C18:0-GlcCer (rs = 0.354; p = 0.018), as well as the sum- GlcCer (rs = 0.312; p = 0.038) (Fig 4).

In the control group statistically significant positive correlations were observed between α-Syn dimer/monomer ratio and C16:0-GlcCer (rs = 0.302; p = 0.035), C18:0- GlcCer (rs = 0.307; p = 0.032), as well as the sum-GlcCer (rs = 0.349; p = 0.014). Additionally, a statistically significant positive correlation was found between a-Syn dimer levels and sum-GlcCer (rs = 0.321; p = 0.025) (Fig 5).

## Discussion

Mutations in the glucocerebrosidase gene (*GBA1*) encoding the lysosomal enzyme GCase are well established as the main genetic risk factor for the development of PD and a plethora of evidence suggests that disturbed a-Syn homeostasis is the link between GD and PD [13, 17–23]. However, the pathogenic mechanism linking these entities is still a topic of debate and both gain- and loss-of-function theories have been put forward, which however are not mutually exclusive [22, 23, 26, 39].

RBCs are rich in α-Syn and in their mature state are devoid of GCase and lysosomes, providing thus an ideal model for the study of α-Syn homeostasis in relation to lipid abnormalities in the absence of lysosomes and of a direct interaction of α-Syn with GCase [27, 40].

Increased oligomerization of α-Syn in RBC membranes of patients with GD has been shown by us and other investigators. However, there is less agreement on the effect of

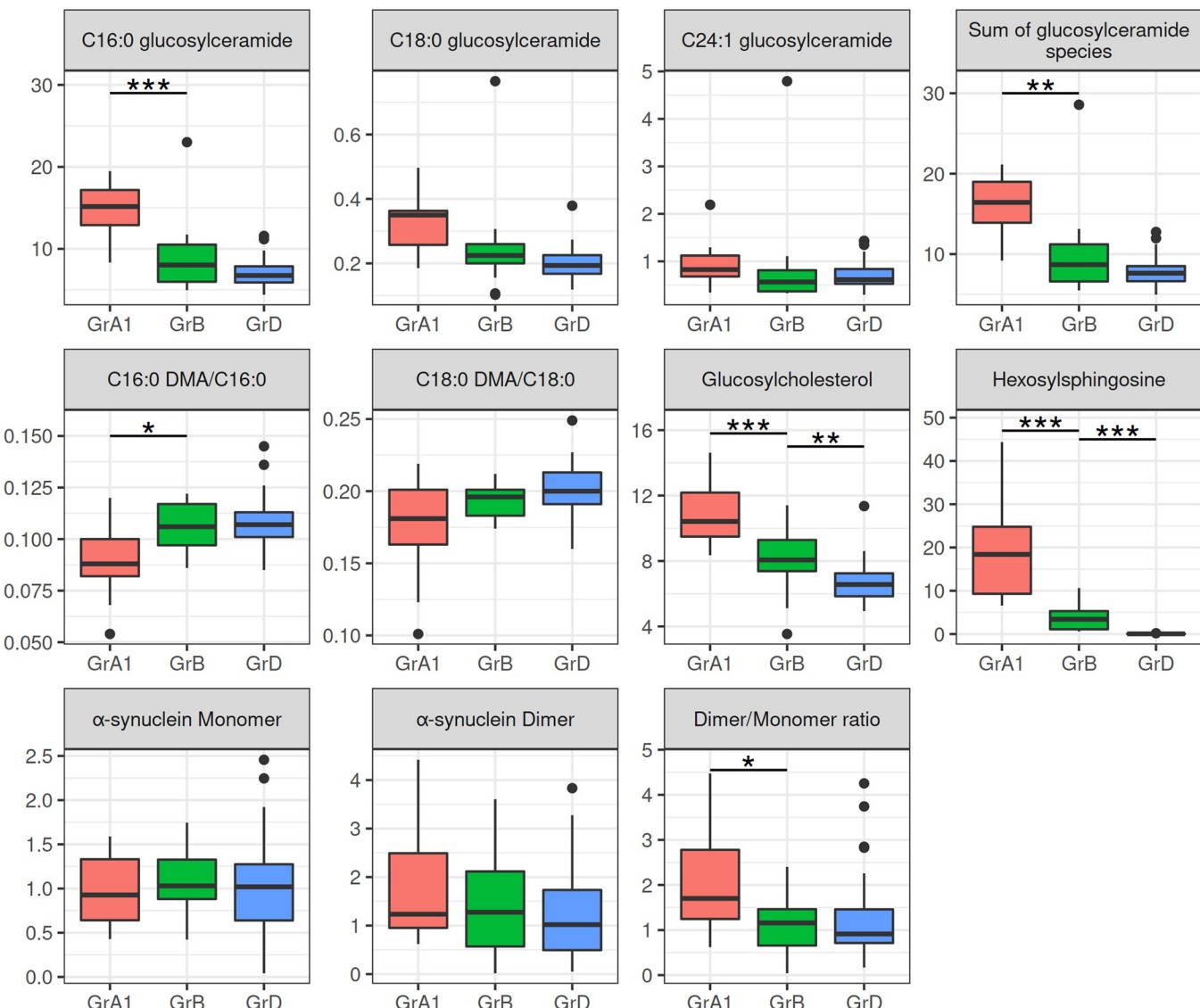

**Fig 3. Red blood cell membrane levels and statistical comparison of the lipids and of α-Synuclein species studied in Gaucher disease patients before and after ERT.** C16:0-glucosylceramide, N-palmitoyl-glucosylceramide; C18:0-glucosylceramide, N-stearoyl-glucosylceramide; C24:1-glucosylceramide, N-nervonoyl-glucosylceramide; C16:0 DMA/C16:0, C16:0-plasmalogens; C18:0 DMA/C18:0, C18:0-plasmalogens; GrA1, Gaucher disease patients receiving no treatment; GrB, the GrA1 Gaucher disease patients following one year of ERT; GrD, controls. Y-axis: the C16:0-, C18:0- and C24:1- glucosylceramide species, their sum, hexosylsphingosine and glucosylcholesterol are expressed in pmoles/$10^8$ cells. The C16:0- and C18:0-plasmalogens are expressed as the ratios of C16:0 DMA to C16:0 and C18:0 DMA to C18:0, respectively. The α-synuclein monomer and dimer are expressed as the ratio of monomeric and dimeric α-Syn to GAPDH, respectively. The α-synuclein dimer/monomer ratio is expressed as the ratio of dimeric to monomeric α-Syn normalized against a reference sample, the dimer to monomer ratio of which was set as 1. *p<0.050, **p<0.010, ***p<0.001. Data were analyzed using Wilcoxon's signed-rank test and Kruskal-Wallis test, with Dunn's post hoc test.

heterozygous *GBA1* mutations (in the presence or absence of PD) on α-Syn behavior [28–34, 41, 42]. In the present study we confirmed our previous results in GD patients [31, 32]. Furthermore, we showed for the first time a significant increase in the α-Syn dimer/monomer ratio in RBC membranes of GD carriers. Since this increase was not associated with an increase in α-Syn monomers, it suggests an increased tendency of conversion of the monomeric to the dimeric form. This indicates that mutations in the *GBA1* gene, even in the heterozygous carrier state, could be a disruptive factor for α-Syn homeostasis and could thus through

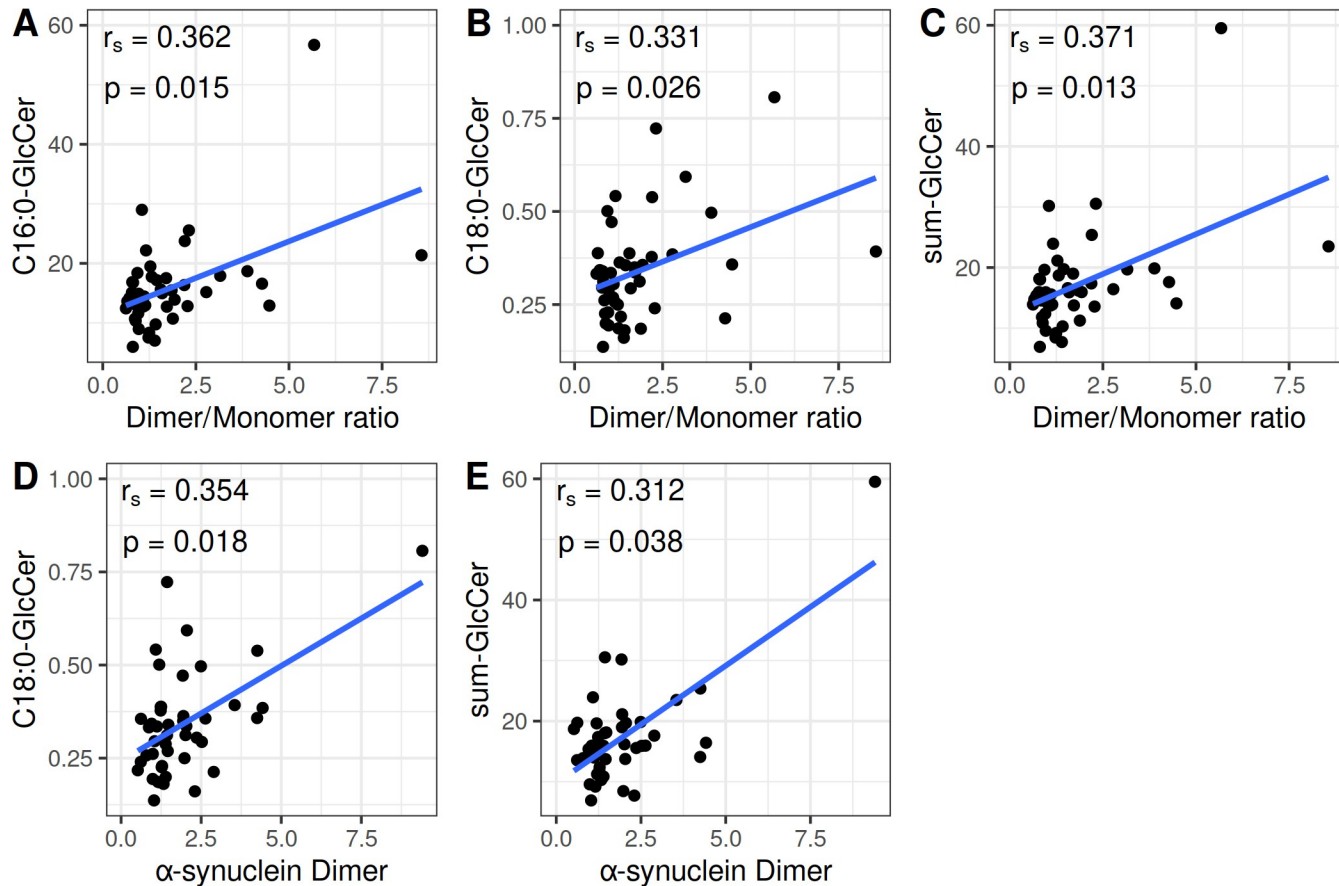

**Fig 4. Statistically significant correlations between the α-Synuclein species and the lipids studied in Gaucher disease patients receiving no treatment.** Correlation between (A) the α-Synuclein dimer/monomer ratio and N-palmitoyl-glucosylceramide (C16:0-GlcCer), (B) the α-Synuclein dimer/monomer ratio and N-stearoyl-glucosylceramide (C18:0-GlcCer), (C) the α-Synuclein dimer/monomer ratio and the sum of glucosylceramide species (sum-GlcCer), (D) the α-Synuclein dimer and N-stearoyl-glucosylceramide (C18:0-GlcCer), (E) the α-Synuclein dimer and the sum of glucosylceramide species (sum-GlcCer). C16:0-GlcCer, C18:0-GlcCer and sum-GlcCer are expressed in pmoles/$10^8$cells. The α-synuclein dimer is expressed as the ratio of dimeric α-Syn to GAPDH. The α-synuclein dimer/monomer ratio is expressed as the ratio of dimeric to monomeric α-Syn.

this mechanism act as a predisposing factor for PD development, as has been shown in multiple genetic studies [13–16, 18].

a-Syn is a lipid binding protein and several studies have shown that membranes and micelles can modulate its aggregation [43]. In vitro studies have shown that vesicles containing GlcCer, GlcSph, sphingosine (Sph) and sphingosine-1-phosphate (S1P), the lipids accumulating in GD, induce changes in the secondary structure of α-Syn and the formation of oligomeric species and amyloid fibrils at acidic pH [43, 44]. We have shown that the increased tendency of α-Syn to form dimers in the RBC membrane of patients with GD correlates with GlcCer levels [32]. The fluidity and curvature of membranes seem to determine the binding affinity of α-Syn, whereas the aggregation of α-Syn is enhanced in the presence of lipids with short saturated hydrocarbon chains [8, 45, 46]. With this in mind, in the present study we extended our previous search to include the study of the different GlcCer species and of HexSph (on average 90% GlcSph in our system) proposed to play a critical role in α-Syn aggregation [44]. Furthermore, we also studied the recently identified GlcChol which is formed by transglucosylation through β-glucosidases using GlcCer as donor and has been found in increased levels in the plasma of GD type I patients [47].

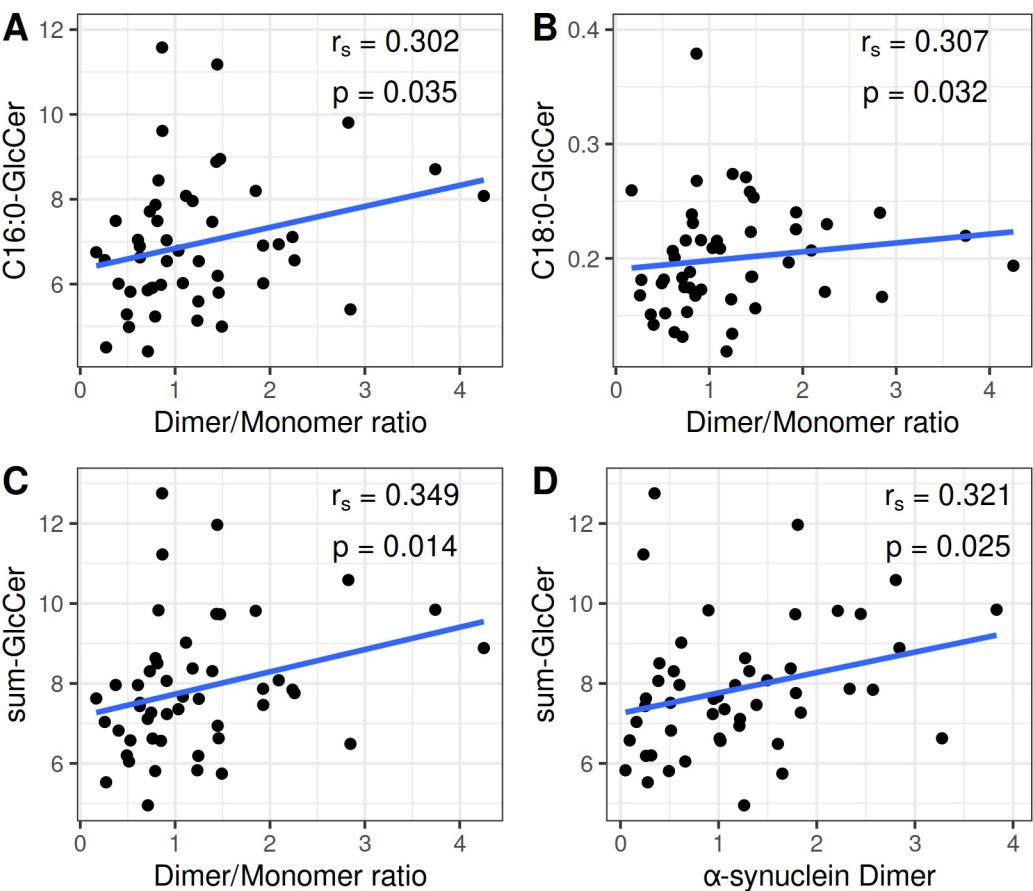

**Fig 5. Statistically significant correlations between the α-Synuclein species and the lipids studied in controls.**
Correlation between (A) the α-Synuclein dimer/monomer ratio and N-palmitoyl-glucosylceramide (C16:0-GlcCer), (B) the α-Synuclein dimer/monomer ratio and N-stearoyl-glucosylceramide (C18:0-GlcCer), (C) the α-Synuclein dimer/monomer ratio and the sum of glucosylceramide species (sum-GlcCer), (D) the α-Synuclein dimer and the sum of glucosylceramide species (sum-GlcCer). C16:0-GlcCer, C18:0-GlcCer and sum-GlcCer are expressed in pmoles/$10^8$cells. The α-synuclein dimer is expressed as the ratio of dimeric α-Syn to GAPDH. The α-synuclein dimer/monomer ratio is expressed as the ratio of dimeric to monomeric α-Syn.

The increase in GlcChol is consistent with these earlier findings, demonstrating degradation of GlcChol by GBA1 and elevated plasma GlcChol in GD patients; increased erythrocyte GlcChol could stem from reduced degradation in the precursor cells, as well as increased exchange in plasma GlcChol.

Plasmalogens represent a unique class of phospholipids, characterized by the presence of a vinyl-ether bond at the sn-1 position of the glycerol backbone. The decrease in their levels observed in GD patients is consistent with our earlier studies [3] and reduced plasmalogen levels could result from the increased oxidative stress observed in GD [48].

We did not find any qualitative differences in the lipids identified in the groups studied. No changes were observed in the profiles of specific lipids analysed, as caused by fatty acid variability, and the classes of lipids identified did not differ markedly. Interestingly the major GlcCer species identified in all the groups studied was C16:0-GlcCer, the species with the shortest acyl chain. Importantly, corroborating and extending our previous results, α-Syn dimer formation in GD patients and controls correlated with the C16:0- and the C18:0-GlcCer species, but not with the C24:1-GlcCer species, supporting a role of short chain acyl moieties in α-Syn oligomerization.

Elevated levels of GlcCer compared to control cells have been identified in heterozygous *GBA1* mutant iPSC-derived neurons [23]. In another study, involving GBA-N370S Parkinson's iPSC-Derived Dopamine Neurons, although no total GlcCer accumulation was observed in patient cells, a marked difference in the GlcCer species was observed compared to controls, with a 30% reduction for C20:0-GlcCer and a 65% increase of C16:0- and C24:0-GlcCer species [49]. In our study no quantitative difference in any of the lipids studied was observed in RBC membranes of GD carriers compared to controls. However, this does not exclude the possibility that local changes in lipid rafts, that could not be detected by our experimental approach, could mediate the oligomerization of α-Syn in *GBA1* mutation carriers.

Importantly, we also show here that ERT had a positive effect on the lipid and α-Syn abnormalities observed in GD patients providing for the first time evidence in human subjects that the homeostatic dysregulation of α-Syn is reversible. This reversion, together with the significant correlations observed between GlcCer and α-Syn dimer, further supports the link between GD and disturbed α-Syn homeostasis.

Although mature erythrocytes lack GCase, the enzyme is expressed and is active in erythroid progenitors. It was suggested that GlcCer accumulation could occur in the early stages of erythropoiesis [50, 51]. In a later study it was shown that GlcCer accumulation in RBCs occurs exclusively during the erythropoiesis process whereas GlcSph, Sph and S1P accumulate both during the erythropoiesis process and in mature RBCs by passive diffusion between the cells and plasma [52]. a-Syn is expressed both in more than 90% of mature peripheral RBCs and also in 80% of erythroid cells in bone marrow including erythroblasts, reticulocytes and erythrocytes [27]. Taken together, the above indicate that α-Syn coexists both with GCase, the lipids associated with GD and lysosomes in the early stages of erythropoiesis. Mazzulli et al [22] have described a pathogenic bidirectional loop between α-syn and GCase depletion in the lysosome. According to their findings, lysosomal GlcCer accumulation accelerates and stabilizes soluble α-syn oligomers. In turn, α-syn accumulation blocks the trafficking of GCase to lysosomes which further amplifies GlcCer accumulation and stabilization of soluble α-syn oligomers resulting in a stronger inhibition of GCase trafficking with each pathogenic cycle. Furthermore, the abnormal conformation of mutant GCase could also contribute by increasing α-synuclein aggregation [53]. Both the above mechanisms could be operating in the early stages of erythropoiesis, where all players are present leading to the oligomerization of α-Syn when *GBA1* mutations are present either in homozygosity or heterozygosity. In conclusion then, we have shown here that in RBC membranes α-Syn oligomerization occurs not only in GD patients but also in carriers of the disorder, even in the absence of PD. In GD patients this oligomerization was associated with lipid abnormalities, whereas no such abnormalities were observed in GD carriers. ERT reverses the biochemical defects and normalizes α-Syn homeostasis, suggesting that, if such replacement were to occur within the brain, it could achieve a therapeutic effect. Further studies investigating α-Syn status during the differentiation of erythroid progenitors could provide new data on the pathogenic mechanism of α-Syn oligomerization in this system.

## Supporting information

**S1 Table. Genotypes of the Gaucher disease patients and carriers studied.**
(DOCX)

**S2 Table. Percentage of Galactosylsphingosine (GalSph) and Glucosylsphingosine (GlcSph) detected following the separation of the Glucose- from the Galactose-containing Hexosylsphingosine (HexSph) species using an HILIC column.**
(DOCX)

**S3 Table. Red blood cell membrane levels and statistical comparison of the lipids studied in Gaucher disease patients receiving no treatment, Gaucher disease carriers and controls.**
(DOCX)

**S4 Table. Red blood cell membrane levels and statistical comparison of the α-synuclein species studied in Gaucher disease patients receiving no treatment, Gaucher disease carriers and controls.**
(DOCX)

**S5 Table. Red blood cell membrane levels and statistical comparison of the lipids studied in Gaucher disease patients before and after ERT.**
(DOCX)

**S6 Table. Red blood cell membrane levels and statistical comparison of the α-synuclein species studied in Gaucher disease patients before and after ERT.**
(DOCX)

## Author Contributions

**Conceptualization:** Helen Michelakakis.

**Data curation:** Marina Moraitou, Nikolaos Papagiannakis.

**Formal analysis:** Nikolaos Papagiannakis.

**Investigation:** Marina Moraitou, Georgios Sotiroudis, Nikolaos Papagiannakis, Maria M. J. Ferraz, Johannes M. F. G. Aerts, Helen Michelakakis.

**Methodology:** Georgios Sotiroudis, Leonidas Stefanis, Helen Michelakakis.

**Project administration:** Leonidas Stefanis, Helen Michelakakis.

**Resources:** Marina Moraitou, Aristotelis Xenakis, Leonidas Stefanis, Helen Michelakakis.

**Supervision:** Aristotelis Xenakis, Johannes M. F. G. Aerts, Leonidas Stefanis, Helen Michelakakis.

**Validation:** Marina Moraitou, Maria M. J. Ferraz, Johannes M. F. G. Aerts, Helen Michelakakis.

**Writing – original draft:** Helen Michelakakis.

**Writing – review & editing:** Marina Moraitou, Georgios Sotiroudis, Nikolaos Papagiannakis, Maria M. J. Ferraz, Johannes M. F. G. Aerts, Leonidas Stefanis, Helen Michelakakis.

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
