## [Decision Letter · Decision Letter 0]

18 Jul 2022

PONE-D-22-16405a-Synuclein and lipids in erythrocytes of Gaucher disease carriers and patients before and after enzyme replacement therapyPLOS ONE

Dear Dr. Michelakakis,

Thank you for submitting your manuscript to PLOS ONE. After careful consideration, we feel that it has merit but does not fully meet PLOS ONE’s publication criteria as it currently stands. Therefore, we invite you to submit a revised version of the manuscript that addresses the points raised during the review process.

We look forward to receiving your revised manuscript.

Kind regards,

David Chau

Academic Editor

PLOS ONE

Journal Requirements:

Reviewers' comments:

Reviewer's Responses to Questions

**Comments to the Author**

1. Is the manuscript technically sound, and do the data support the conclusions?

Reviewer #1: Partly

Reviewer #2: Partly

2. Has the statistical analysis been performed appropriately and rigorously? 

Reviewer #1: Yes

Reviewer #2: Yes

3. Have the authors made all data underlying the findings in their manuscript fully available?

Reviewer #1: Yes

Reviewer #2: Yes

4. Is the manuscript presented in an intelligible fashion and written in standard English?

Reviewer #1: Yes

Reviewer #2: Yes

5. Review Comments to the Author

Reviewer #1: This study analyzed the lipids and alpha-synuclein extracted from erythrocyte membrane of patients and carriers with GBA1 variants. The effects of enzyme replacement therapy on lipids and alpha-synuclein aggregation were also analyzed, which would be valuable for the related research field. However, there are lacks of detail in clinical data and validation of the assay methods, which need to be filled in.

Major points:

Please provide a breakdown of the racial groups analyzed. For groups with disease variants, please provide a table with a breakdown of each variant.

Please indicate a representative blot pattern of alpha-synuclein.

Please provide a figure representing the correlations between significant lipid species and alpha-synuclein.

Were the authors able to distinguish between GlcChol and GalChol and between GlcSph and GalSph using the methods described in the manuscript? If so, please provide the methods in detail, including the validation data. If not, change the notations to HexChol and HexSph, or provide data showing that GalChol and GalSph were barely present in the erythrocyte membrane.

Minor points:

It would be better if the mean corpuscular volume of erythrocytes were presented.

In the introduction, MIM should be spelled out for the first time.

Although the binding of alpha-synuclein to the membrane is transient, the method of extraction of alpha-synuclein from the erythrocyte membrane is an important factor in the interpretation of the results. Please provide details rather than citing the non-open access literature of the authors.

For the sentence “We did not find any qualitative differences in the lipids identified in the groups studied.” (line 308), given the diversity of lipid species present in erythrocyte membrane, it seems that the lipid molecular species the authors analyzed are quite limited. Please describe exactly what the authors mean by “qualitative differences”.

Please discuss the results showing an increase in GlcCho and a decrease in plasmalogens by GBA1 mutations.

The data on alpha-synuclein ratio in GrD (the range of age is 4-76 yrs), has a high dispersion. Did the authors analyze the correlation of the ratio with age?

Reviewer #2: In the presented manuscript the authors summarized their results on specific lipid content and on the amount of monomeric versus dimeric α-Synuclein in membranes of red blood cells. The red blood cells originated from Gaucher disease (GD) patients, from GD patients before and after a year of enzyme replacement therapy, from carriers of mutant GBA1 and from control individuals. The authors showed that:

* There was Increased oligomerization of α-Synuclein in red blood cell membranes in GD patients

and in carriers of GBA1 mutations. There was no significant difference in oligomerization between these two groups.

* There were significant differences (which did not exceed three-fold) in the lipids studied between Gaucher disease patients and controls but not between carriers of GBA1 mutations compared to controls. Enzyme Replacement Therapy (ERT) reversed the defects in the content of the lipids studied and normalized α-Synuclein homeostasis in red blood cells.

The results are nicely presented and discussed, and the manuscript deserves publication in PlosONE.

There several points that need the authors’ attention, as follows:

1. The authors presented interesting results on lipid content and on α-Synuclein monomer/dimer ratios in red blood cells membranes that originated from GD patients, from carriers of GBA1 mutations and from controls. Since none of the individuals studied developed Parkinson disease (PD), the relevance of the presented results to GBA1 associated PD, and the discussion on this subject, is not clear.

2. Introduction, lines 80-81: the authors mention gain-of-function and loss-of-function theories. The fact that mutant GBA1 carriers are at risk to develop PD strongly indicates it is a dominant trait, for which gain-of-function or haploinsufficiency apply. Loss-of-function is reserved for recessive traits.

3. Subjects, Lines 107-110: Subjects were divided into four different groups. In group A there where 13 patients before (group A1) and after one year of treatment (group B). Does this mean that the remaining 32 patients in group A have not been treated? Please add this detail to the text.

4. Methods, lines 188-197: The authors describe western blotting to detect α-Synuclein monomers and dimers. It would have been advantageous to present such a blot in the results section.

5. Figures 1, 2: It is not clear what the Y-axes present in the different panels.

6. Discussion, lines 291-294: The authors argue that: “This indicates that mutations in the GBA1 gene, even in the heterozygous carrier state, could be a disruptive factor for α-Syn homeostasis and could thus through this mechanism act as a predisposing factor for PD development, as has been shown in multiple genetic studies [13-16, 18].” It is difficult to accept the explanation presented by the authors for mature red blood cells. Did the authors consider the possibility that in the absence of lysosomes, with no disposal of a-Synuclein molecules they accumulate and eventually aggregate?

7. Discussion, lines 335-339: I have difficulties accepting the conclusion that: “Taken together, the above indicate that α-Syn coexists both with GCase and the lipids associated with GD in the early stages of erythropoiesis. This coexistence could result in the formation of the pathogenic bidirectional loop described by Mazzulli et al [22] leading to the oligomerization of α-Syn when GBA1 mutations are present either in homozygosity or heterozygosity.” The authors should explain what does “The pathogenic bidirectional loop described by Mazzulli “ mean.

8. Discussion, lines 340-343: The authors claim that “we have shown here that in red blood cells’ membranes α-Syn oligomerization occurs not only in GD patients but also in carriers of the disorder, even in the absence of PD. In GD patients this oligomerization was associated with lipid abnormalities, whereas no such abnormalities were observed in GD carriers.” So, what is the reason for α-Synuclein accumulation in carriers of mutant GBA1?

9. Table S2: The authors claim that there is increased oligomerization of α-Synuclein in red blood cell membranes in Gaucher disease patients and in carriers of the disease. I did not see increased dimerization in carriers.

6. PLOS authors have the option to publish the peer review history of their article (what does this mean?). If published, this will include your full peer review and any attached files.

Reviewer #1: **Yes: **Yuzuru Imai

Reviewer #2: No

---

## [Author Response · Author response to Decision Letter 0]

5 Oct 2022

Editor comment:Please provide additional details regarding participant consent. In the ethics statement in the Methods and online submission information, please ensure that you have specified what type you obtained (for instance, written or verbal, and if verbal, how it was documented and witnessed). If your study included minors, state whether you obtained consent from parents or guardians. If the need for consent was waived by the ethics committee, please include this information.

Lines 116-117 Subjects the following was added: The need for consent was waived by the ethics committee.

Reviewer #1: Please provide a breakdown of the racial groups analyzed. For groups with disease variants, please provide a table with a breakdown of each variant.

Line 105 Subjects becomes : A total of 113 individuals all of Caucasian origin were included in this study. 

S1 Table with the genotypes of the GD patients and carriers studied was added and 

Line 113-114 Subjects becomes: The genotypes of the GD patients and carriers studied are shown in S1 Table.

Please indicate a representative blot pattern of alpha-synuclein.

Thank you for your suggestion. The data were added as Fig 1. 

Lines 244-245 now read: Representative immunoblots of α-synuclein monomer and dimer for the four different study groups are shown in Fig 1.

Lines 246-250 now read: Fig 1. Representative immunoblots of α-synuclein monomer and dimer for the four different study groups. GrA, Gaucher disease patients receiving no treatment; GrB, Gaucher disease patients after treatment; GrC, Gaucher disease carriers; GrD, controls. Quantification was made relative to the levels of the loading control (GAPDH, glyceraldehyde 3-phosphate dehydrogenase).

Please provide a figure representing the correlations between significant lipid species and alpha-synuclein.

Thank you for your suggestion.

The correlations were added as Fig 4 (Line 319) and Fig 5 (line 334).

The legends were added in the Results section. 

Lines 320-329 now read: Fig 4. Statistically significant correlations between the α-Synuclein species and the lipids studied in Gaucher disease patients receiving no treatment. Correlation between (A) the α-Synuclein dimer/monomer ratio and N-palmitoyl-glucosylceramide (C16:0-GlcCer), (B) the α-Synuclein dimer/monomer ratio and N-stearoyl-glucosylceramide (C18:0-GlcCer), (C) the α-Synuclein dimer/monomer ratio and the sum of glucosylceramide species (sum-GlcCer), (D) the α-Synuclein dimer and N-stearoyl-glucosylceramide (C18:0-GlcCer), (E) the α-Synuclein dimer and the sum of glucosylceramide species (sum-GlcCer). C16:0-GlcCer, C18:0-GlcCer and sum-GlcCer are expressed in pmoles/108cells. The α-synuclein dimer is expressed as the ratio of dimeric α-Syn to GAPDH. The α-synuclein dimer/monomer ratio is expressed as the ratio of dimeric to monomeric α-Syn.

Lines 335-343 now read : Fig 5. Statistically significant correlations between the α-Synuclein species and the lipids studied in controls. Correlation between (A) the α-Synuclein dimer/monomer ratio and N-palmitoyl-glucosylceramide (C16:0-GlcCer), (B) the α-Synuclein dimer/monomer ratio and N-stearoyl-glucosylceramide (C18:0-GlcCer), (C) the α-Synuclein dimer/monomer ratio and the sum of glucosylceramide species (sum-GlcCer), (D) the α-Synuclein dimer and the sum of glucosylceramide species (sum-GlcCer). C16:0-GlcCer, C18:0-GlcCer and sum-GlcCer are expressed in pmoles/108cells. The α-synuclein dimer is expressed as the ratio of dimeric α-Syn to GAPDH. The α-synuclein dimer/monomer ratio is expressed as the ratio of dimeric to monomeric α-Syn.

Were the authors able to distinguish between GlcChol and GalChol and between GlcSph and GalSph using the methods described in the manuscript? If so, please provide the methods in detail, including the validation data. If not, change the notations to HexChol and HexSph, or provide data showing that GalChol and GalSph were barely present in the erythrocyte membrane.

Thank you for your question.

Samples were run using a HILIC column in order to separate between Galactose and Glucose containing molecules. 

According to our findings no GalChol could not be detected, so the term GlcChol was not changed in the text. HexChol was only used in the methods section Lines 168 and 205

On the other hand, on average 9% galactosylsphingosine vs. 91% glucosylsphingosine was detected. Following this finding GlcSph was changed to HexSph. Data are included in S2 Table

The following changes were made in the text:

Lines 98-99 Introduction become: 

hexosylsphingosine (HexSph), being largely glucosylsphingosine (GlcSph),

Lines 167-169 Methods become: The quantification of glycosylated sphingosine (HexSph, the total of glucosylsphingosine and galactosylsphingosine) and of glycosylated cholesterol (HexChol, the total of glucosylcholesterol (GlcChol) and galactosylcholesterol (GalChol) 

Lines 192-209 Methods the following was added:

In order to separate galactose- from glucose-containing molecules, samples were run in an ethylene bridged hybrid (BEH) hydrophilic interaction liquid chromatography (HILIC) column as previously described [38]. Glycosylated sphingosine and glycosylated cholesterol were extracted using the acidic Bligh and Dyer procedure previously described, with lipids resuspended in acetonitrile:methanol (9:1, v/v) prior to transfer to LC-MS vials. LC-MS/MS measurements were performed using a Waters UPLC-Xevo-TQS micro instrument (Waters, Corporation, Milford, USA) in positive mode using an ESI source. A BEH HILIC column (2.1 × 100 mm with 1.7 μm particle size, Waters) was used at 30°C as described before [38]. Eluent A contained 10 mM ammonium formate in acetonitrile/water (97:3, v/v) with 0.01% (v/v) formic acid, and eluent B consisted of 10 mM ammonium formate in acetonitrile/water (75:25, v/v) with 0.01% (v/v) formic acid. HexSph was eluted in 18 min with a flow of 0.4 ml/min using the following program: 85% A from 0 to 2 min, 85–70% A from 2 to 2.5 min, 70% A from 2.5 to 5.5 min, 70–60% A from 5.5 to 6 min, 60% A from 6 to 8 min, 60–0% A from 8 to 8.5 min, 0–85% A from 8.5 to 9.5 min, and re-equilibration of the column with 85% A from 10 to 18 min. HexChol was eluted in 18 min with a flow of 0.25 ml/min using the following program: 100% A from 0 to 3 min, 100–0% A from 3 to 3.5 min, 0% A from 3.5 to 4.5 min, 0–100% A from 4.5 to 5 min, and re-equilibration with 100% A from 5 to 18 min. Data were analysed with MassLynx 4.1 software (Waters Corporation).

The following reference was added in references as No 38:

Lelieveld LT, Mirzaian M, Kuo CL, Artola M, Ferraz MJ, Peter REA, et al. Role of β-glucosidase 2 in aberrant glycosphingolipid metabolism: model of glucocerebrosidase deficiency in zebrafish. J Lipid Res. 2019;60:1851-1867. doi: 10.1194/jlr.RA119000154. 

Line 241-243 Results the following was added: The use of a HILIC column allows the separation of glucosyl- and galactosyl-lipids. Analysis of six samples revealed that in the samples studied, the HexSph was on average 90% GlcSph (S2 Table). On the other hand, no galactosylcholesterol was detected in any of the samples. 

Lines 377-378 Discussion the following was added: HexSph (on average 90% GlcSph in our system)

Minor points:

It would be better if the mean corpuscular volume of erythrocytes were presented.

Thank you for your question. Counting of red blood cells was performed not in the native blood sample but after the red blood cell pellet had been washed twice and resuspended in 0.9% saline. We believe that this treatment could have an effect on the status of red blood cells giving erroneous results if MCV was measured and used for expressing the results.

In the introduction, MIM should be spelled out for the first time.

Line 47 the following was added: Mendelian Inheritance in Man

Although the binding of alpha-synuclein to the membrane is transient, the method of extraction of alpha-synuclein from the erythrocyte membrane is an important factor in the interpretation of the results. Please provide details rather than citing the non-open access literature of the authors.

Thank you for your comment

Lines 210-221 Methods now read:

Levels of monomeric and dimeric α-Syn in RBC membranes were assessed using Western Immunoblotting. Red blood cell membrane pellets were purified from lysed RBCs through repeated washes in cold PBS and centrifugation at 1,000 g for 10 min at 4 ◦C. Supernatants containing cytosolic proteins, mainly hemoglobin, were discarded. The membrane pellets were then solubilized with 50 μl STET lysis buffer (50 mM Tris (pH 7.6), 150 mM NaCl, 2 mM EDTA, 1% Triton-X) on ice for 30 min as described in Argyriou et al [31]. Forty micrograms of protein were run on 8% sodium dodecyl sulfate-polyacrylamide gels. Βlots were probed with the antibodies directed against α-synuclein (C-20 polyclonal antibody, sc-7011-R, Santa Cruz Biotechnology) and glyceraldehyde 3-phosphate dehydrogenase (GAPDH) (AB2302, Millipore). The ratio of monomeric or dimeric α-synuclein to GAPDH, used as loading control, was first assessed. To make comparisons between different experiments, an internal control was always included on each blot as the reference point [29]. 

 For the sentence “We did not find any qualitative differences in the lipids identified in the groups studied.” (line 308), given the diversity of lipid species present in erythrocyte membrane, it seems that the lipid molecular species the authors analyzed are quite limited. Please describe exactly what the authors mean by “qualitative differences”.

We agree that our study did not cover all the lipids present in RBC membranes which in any case was beyond the scope of the study. By the term “qualitative differences” we refer to the finding that no changes were observed in the profiles of specific lipids analyzed as caused by fatty acid variability. Furthermore the classes of lipids identified did not differ markedly. 

The following was added:

Lines 390-392 Discussion: No changes were observed in the profiles of specific lipids analyzed, as caused by fatty acid variability, and the classes of lipids identified did not differ markedly.

Please discuss the results showing an increase in GlcCho and a decrease in plasmalogens by GBA1 mutations.

Thank you for your comment.

The following were added in the discussion:

Lines 382- 385: The increase in GlcChol is consistent with these earlier findings demonstrating degradation of GlcChol by GBA1 and elevated plasma GlcChol in GD patients; increased erythrocyte GlcChol could stem from reduced degradation in the precursor cells as well as increased exchange in plasma GlcChol.

Lines 386- 389 : Plasmalogens represent a unique class of phospholipids, characterized by the presence of a vinyl-ether bond at the sn-1 position of the glycerol backbone. The decrease in their levels is consistent with our earlier studies [3] and reduced plasmalogen levels could be the result of increased oxidative stress as indicated by previous studies [48].

Furthermore the paper: Moraitou M, Dimitriou E, Dekker N, Monopolis I, Aerts J, Michelakakis H. Gaucher disease: plasmalogen levels in relation to primary lipid abnormalities and oxidative stress. Blood Cells Mol Dis. 2014; 53:30-33. doi: 10.1016/j.bcmd.2014.01.005. was added as ref 48.

The data on alpha-synuclein ratio in GrD (the range of age is 4-76 yrs), has a high dispersion. Did the authors analyze the correlation of the ratio with age?

We did the analysis and no correlation was observed between the α-Syn dimer to monomer ratio and age (rs=-0.127, p=0.383).

Reviewer #2: 1. The authors presented interesting results on lipid content and on α-Synuclein monomer/dimer ratios in red blood cells membranes that originated from GD patients, from carriers of GBA1 mutations and from controls. Since none of the individuals studied developed Parkinson disease (PD), the relevance of the presented results to GBA1 associated PD, and the discussion on this subject, is not clear.

Thank you for your comment. Several studies have pointed to disturbed α-Syn homeostasis as the link between PD and GD. The present study was designed to investigate further the disturbed α-Syn homeostasis already observed in GD in relation to lipid abnormalities in carriers and the response to treatment. GBA1 mutations are a proven risk factor for developing PD and our results show that disturbed α-syn homeostasis could be a predisposing factor. However how and why this, in some individuals will translate into development of PD, remains an enigma which our study was not aiming in answering. 

2. Introduction, lines 80-81: the authors mention gain-of-function and loss-of-function theories. The fact that mutant GBA1 carriers are at risk to develop PD strongly indicates it is a dominant trait, for which gain-of-function or haploinsufficiency apply. Loss-of-function is reserved for recessive traits.

Thank you for your comment. We agree that loss-of-function is commonly associated with recessive traits. In lines 80-81 we refer to the arguments raised for gain- and loss-of-function theories that have been put forward in discussing the possible pathogenic mechanisms linking PD and GD. 

3. Subjects, Lines 107-110: Subjects were divided into four different groups. In group A there where 13 patients before (group A1) and after one year of treatment (group B). Does this mean that the remaining 32 patients in group A have not been treated? Please add this detail to the text.

Thank you for your question.

Line 110-111 Subjects the following was added:

The remaining 32 patients in group A either were not treated or there was no blood sample available after one year of treatment.

4. Methods, lines 188-197: The authors describe western blotting to detect α-Synuclein monomers and dimers. It would have been advantageous to present such a blot in the results section.

Thank you very much. 

The Blot has been included as Fig 1. 

Furthermore, the following was added:

Lines 246-250 : Fig 1. Representative immunoblots of α-synuclein monomer and dimer for the four different study groups. GrA, Gaucher disease patients receiving no treatment; GrB, Gaucher disease patients after treatment; GrC, Gaucher disease carriers; GrD, controls. Quantification was made relative to the levels of the loading control (GAPDH, glyceraldehyde 3-phosphate dehydrogenase).

5. Figures 1, 2: It is not clear what the Y-axes present in the different panels.

The following was added in the legends of Fig 1,2 (now Fig 2,3 following the addition of the immunoblot):

Lines 270-275 and Lines 306-311: Y-axis: the C16:0-, C18:0- and C24:1- glucosylceramide species, their sum, hexosylsphingosine and glucosylcholesterol are expressed in pmoles/108cells. The C16:0- and C18:0-plasmalogens are expressed as the ratios of C16:0 DMA to C16:0 and C18:0 DMA to C18:0, respectively. The α-synuclein monomer and dimer are expressed as the ratio of monomeric and dimeric α-Syn to GAPDH, respectively. The α-synuclein dimer/monomer ratio is expressed as the ratio of dimeric to monomeric α-Syn.

6. Discussion, lines 291-294: The authors argue that: “This indicates that mutations in the GBA1 gene, even in the heterozygous carrier state, could be a disruptive factor for α-Syn homeostasis and could thus through this mechanism act as a predisposing factor for PD development, as has been shown in multiple genetic studies [13-16, 18].” It is difficult to accept the explanation presented by the authors for mature red blood cells. Did the authors consider the possibility that in the absence of lysosomes, with no disposal of a-Synuclein molecules they accumulate and eventually aggregate?

Thank you for your comment. If the mere lack of lysosomes in mature red blood cells was the cause of the disturbed homeostasis of α-Syn, it is reasonable to think that this should be also detected in mature red blood cells of controls and possibly, as a mechanism, could affect other proteins as well. Our hypothesis is that α-Syn status/properties are already disturbed before the red blood cells become mature. 

7. Discussion, lines 335-339: I have difficulties accepting the conclusion that: “Taken together, the above indicate that α-Syn coexists both with GCase and the lipids associated with GD in the early stages of erythropoiesis. This coexistence could result in the formation of the pathogenic bidirectional loop described by Mazzulli et al [22] leading to the oligomerization of α-Syn when GBA1 mutations are present either in homozygosity or heterozygosity.” The authors should explain what does “The pathogenic bidirectional loop described by Mazzulli “ mean.

Taken together, the above indicate that α-Syn coexists both with GCase the lipids associated with GD in the early stages of erythropoiesis. This coexistence could result in the formation of the pathogenic bidirectional loop described by Mazzulli et al [22] leading to the oligomerization of α-Syn when GBA1 mutations are present either in homozygosity or heterozygosity. 

Thank you for your comment.

The following changes have been made:

Lines 419-427 now read:

Taken together, the above indicate that α-Syn coexists both with GCase, the lipids associated with GD and lysosomes in the early stages of erythropoiesis. Mazzulli et al [22] have described a pathogenic bidirectional loop between α-syn and GCase depletion in the lysosome. According to his findings lysosomal GlcCer accumulation accelerates and stabilizes soluble α-syn oligomers. In turn, α-syn accumulation blocks the trafficking of GCase to lysosomes which further amplifies GlcCer accumulation and stabilization of soluble α-syn oligomers resulting in a stronger inhibition of GCase trafficking with each pathogenic cycle. Furthermore, the abnormal confirmation of mutant GCase could also contribute by increasing α-synuclein aggregation [53]. 

Furthermore, the paper by Sidransky E, Lopez G. The link between the GBA gene and parkinsonism. Lancet Neurol. 2012; 11: 986-98. doi: 10.1016/S1474-4422(12)70190-4 was included as Ref.no 53

8. Discussion, lines 340-343: The authors claim that “we have shown here that in red blood cells’ membranes α-Syn oligomerization occurs not only in GD patients but also in carriers of the disorder, even in the absence of PD. In GD patients this oligomerization was associated with lipid abnormalities, whereas no such abnormalities were observed in GD carriers.” So, what is the reason for α-Synuclein accumulation in carriers of mutant GBA1?

Please see above

9. Table S2: The authors claim that there is increased oligomerization of α-Synuclein in red blood cell membranes in Gaucher disease patients and in carriers of the disease. I did not see increased dimerization in carriers.

Thank you for pointing this out

The following changes were made:

Lines 285-290 Results : Although an increase in α-Syn dimer levels was observed in GD carriers (median: 1.32; range: 0.49-3.39) compared to controls (median: 1.02; range: 0.05-3.83) this difference did not reach statistical significance. A statistically significant increase was however observed in the α-Syn dimer/monomer ratio (p=0.019) in GD carriers compared to controls. On the other hand no statistically significant differences were observed between GD patients and carriers (Fig 2, S4 Table).

Lines 366-370 : Furthermore, we showed for the first time a significant increase in the α-Syn dimer/monomer ratio in RBC membranes of GD carriers. Since this increase was not associated with an increase in α-Syn monomers, it suggests an increased tendency of conversion of the monomeric to the dimeric form.

---

## [Decision Letter · Decision Letter 1]

17 Oct 2022

PONE-D-22-16405R1a-Synuclein and lipids in erythrocytes of Gaucher disease carriers and patients before and after enzyme replacement therapyPLOS ONE

Dear Dr. Michelakakis,

Thank you for submitting your manuscript to PLOS ONE. After careful consideration, we feel that it has merit but does not fully meet PLOS ONE’s publication criteria as it currently stands. Therefore, we invite you to submit a revised version of the manuscript that addresses the points raised during the review process.

We look forward to receiving your revised manuscript.

Kind regards,

David Chau

Academic Editor

PLOS ONE

Journal Requirements:

Reviewers' comments:

Reviewer's Responses to Questions

**Comments to the Author**

1. If the authors have adequately addressed your comments raised in a previous round of review and you feel that this manuscript is now acceptable for publication, you may indicate that here to bypass the “Comments to the Author” section, enter your conflict of interest statement in the “Confidential to Editor” section, and submit your "Accept" recommendation.

Reviewer #1: (No Response)

Reviewer #2: All comments have been addressed

2. Is the manuscript technically sound, and do the data support the conclusions?

Reviewer #1: Partly

Reviewer #2: Yes

3. Has the statistical analysis been performed appropriately and rigorously? 

Reviewer #1: Yes

Reviewer #2: Yes

4. Have the authors made all data underlying the findings in their manuscript fully available?

Reviewer #1: Yes

Reviewer #2: Yes

5. Is the manuscript presented in an intelligible fashion and written in standard English?

Reviewer #1: Yes

Reviewer #2: Yes

6. Review Comments to the Author

Reviewer #1: I am approximately satisfied with the authors' responses, but there are two points that need to be corrected.

In the newly added Fig 1, the result does not appear to reflect the graphs in Fig 2. Also, the dimer and monomer bands should be shown in the same blot without cropping.

Regarding genotype in Table S1, please add description of IVS and RecNciI.

Reviewer #2: The authors have adequately addressed my main concerns, and therefore I accept the revised manuscript as is.

7. PLOS authors have the option to publish the peer review history of their article (what does this mean?). If published, this will include your full peer review and any attached files.

Reviewer #1: **Yes: **Yuzuru Imai

Reviewer #2: No

---

## [Author Response · Author response to Decision Letter 1]

31 Oct 2022

Reviewer #1: I am approximately satisfied with the authors' responses, but there are two points that need to be corrected.

In the newly added Fig 1, the result does not appear to reflect the graphs in Fig 2. Also, the dimer and monomer bands should be shown in the same blot without cropping.

Thank you for your comment.

 The results in fig1 and fig2 are not fully comparable, since the samples in the blot were randomly selected from each group. Furthermore, as stated in the relevant method (lines 219-223), the data shown in fig2 result from the calculation of the band intensity ratios between the monomer and GAPDH (a-Syn monomer), the dimer and GAPDH (a-Syn dimer) and the dimer and monomer (dimer/monomer ratio) after normalization.

We have replaced the blot so that the monomer and dimer bands are shown in the same blot without cropping.

Furthermore the following was added in the legend of Fig 2 (line 275) and 3 (line 311): The α-synuclein dimer/monomer ratio is expressed as the ratio of dimeric to monomeric α-Syn normalized against a reference sample, the dimer to monomer ratio of which was set as 1.

Regarding genotype in Table S1, please add description of IVS and RecNciI.

In response to the comment the following was added in the legend of the Table S1

RecNciI (L444P, A456P, V460V) 

IVS10-1G>A (c.[1505+1_1505+12ins;1505G>A])

---

## [Decision Letter · Decision Letter 2]

1 Nov 2022

a-Synuclein and lipids in erythrocytes of Gaucher disease carriers and patients before and after enzyme replacement therapy

PONE-D-22-16405R2

Dear Dr. Michelakakis,

We’re pleased to inform you that your manuscript has been judged scientifically suitable for publication and will be formally accepted for publication once it meets all outstanding technical requirements.

Kind regards,

David Chau

Academic Editor

PLOS ONE

Additional Editor Comments (optional):

Reviewers' comments:

Reviewer's Responses to Questions

**Comments to the Author**

1. If the authors have adequately addressed your comments raised in a previous round of review and you feel that this manuscript is now acceptable for publication, you may indicate that here to bypass the “Comments to the Author” section, enter your conflict of interest statement in the “Confidential to Editor” section, and submit your "Accept" recommendation.

Reviewer #1: All comments have been addressed

2. Is the manuscript technically sound, and do the data support the conclusions?

Reviewer #1: Yes

3. Has the statistical analysis been performed appropriately and rigorously? 

Reviewer #1: Yes

4. Have the authors made all data underlying the findings in their manuscript fully available?

Reviewer #1: Yes

5. Is the manuscript presented in an intelligible fashion and written in standard English?

Reviewer #1: Yes

6. Review Comments to the Author

Reviewer #1: (No Response)

7. PLOS authors have the option to publish the peer review history of their article (what does this mean?). If published, this will include your full peer review and any attached files.

Reviewer #1: **Yes: **Yuzuru Imai

---

## [Editor Report · Acceptance letter]

5 Nov 2022

PONE-D-22-16405R2 

a-Synuclein and lipids in erythrocytes of Gaucher disease carriers and patients before and after enzyme replacement therapy 

Dear Dr. Michelakakis:

I'm pleased to inform you that your manuscript has been deemed suitable for publication in PLOS ONE. Congratulations! Your manuscript is now with our production department. 

Kind regards, 

on behalf of

Dr. David Chau 

Academic Editor

PLOS ONE